# Effective vaccine allocation strategies, balancing economy with infection control against COVID-19 in Japan

**Satoshi Sunohara**[1]☯, **Toshiaki Asakura**[1]☯, **Takashi Kimura**[2]*, **Shun Ozawa**[1], **Satoshi Oshima**[1], **Daigo Yamauchi**[1], **Akiko Tamakoshi**[2]

**1** School of Medicine, Hokkaido University, Sapporo, Japan, **2** Public Health, Faculty of Medicine, Hokkaido University, Sapporo, Japan

☯ These authors contributed equally to this work.

\* kimura@med.hokudai.ac.jp

**Data Availability Statement:** All relevant data are within the manuscript and its Supporting Information files.

## Abstract

Due to COVID-19, many countries including Japan have implemented a suspension of economic activities for infection control. It has contributed to reduce the transmission of COVID-19 but caused severe economic losses. Today, several promising vaccines have been developed and are already being distributed in some countries. Therefore, we evaluated various vaccine and intensive countermeasure strategies with constraint of economic loss using SEIR model to obtain knowledge of how to balance economy with infection control in Japan. Our main results were that the vaccination strategy that prioritized younger generation was better in terms of deaths when a linear relationship between lockdown intensity and acceptable economic loss was assumed. On the other hand, when a non-linearity relationship was introduced, implying that the strong lockdown with small economic loss was possible, the old first strategies were best in the settings of small basic reproduction number. These results indicated a high potential of remote work when prioritizing vaccination for the old generation. When focusing on only the old first strategies as the Japanese government has decided to do, the strategy vaccinating the young next to the old was superior to the others when a non-linear relationship was assumed due to sufficient reduction of contact with small economic loss.

## Introduction

At the end of 2019, an outbreak of coronavirus disease 2019 (COVID-19) caused by the SARS-CoV-2 virus began in Wuhan City (Hubei Province, China) [1, 2]. Over one hundred million people worldwide have been infected with COVID-19 since it was declared to be a pandemic by the World Health Organization on March 11th, 2020. As of April 1st, 2021, the death toll was over 2.8 million worldwide and over 9,000 in Japan [3].

Due to this pandemic, many countries have implemented a suspension of economic activities (usually referred to as a lockdown) with restrictions on movement [4, 5]. It has been observed that the implementation of the lockdown reduces contact rate and thus transmission

**Funding:** The authors received no specific funding for this work.

**Competing interests:** The authors have declared that no competing interests exist.

of SARS-CoV-2 [6–9]. On the other hand, lockdowns stopped economic activity and caused severe economic losses [10–12]. In Japan, no enforceable policy of movement restrictions was implemented, but the government declared a state of emergency, not a strict lockdown, to control the spread of SARS-CoV-2. This declaration allowed prefectural governors to take measures such as restricting the operation of public facilities, including schools, and reducing the hours of operation of restaurants [13, 14]. This measure resulted in economic losses like lockdown policies in other countries. In the situation of pandemic of COVID-19, the world's GDP and Japan's GDP were estimated to have shrunk by 5.2% [15] and by 4.8% [16] in 2020, respectively.

Today, unlike the situation at the time of the emergence of SARS-CoV-2, several promising vaccines have been developed and are already being distributed in some countries [17]. As of the 15th of June, 20.8% of the world's population has been vaccinated at least once, especially in many developed countries, where more than 40% of the population has already been vaccinated. However, in low-income countries, only 0.8% of the population has been vaccinated at least once, and vaccination in low-income countries is still an issue. As a consequence, some developed countries like Israel and UK have confirmed a decrease in the number of newly reported case of infection [3]. In Japan, vaccination for healthcare workers has begun in March 2021, and vaccination for the general elderly population has been implemented from April 12th, 2021. However, until herd immunity is established by vaccinating a certain percentage of the population, it is necessary to continue controlling contacts as a preventive measure against infection. This will require knowledge of how to balance economy with infection control under progression of vaccination.

Therefore, we evaluated various vaccine and intensive countermeasure strategies with constraint of predefined amount of economical loss using SEIR model.

## Materials and methods

### Simulation scenarios

We assumed that vaccine was started to be distributed at the start of the simulation and that all individuals received vaccination within one year at a constant rate. We used a lockdown as one of intensive countermeasures. Acceptable economic loss was predefined and intensity, length and start timing of the lockdown were optimized to minimize the cumulative number of deaths at the end of the simulation, comparing with different vaccination strategies. It is noted that the lockdown used in this study is defined as countermeasures not only reducing transmissibility but also causing economical damage. Our usage of the lockdown includes containment and closure indicators of OxCGRT indicators [13], for example workplace closing, cancel public events, restrictions on gathering size and stay-at-home requirements. On the other hand, neither countermeasures without economical damage (ex. mask wearing, improving hand hygiene and public information campaign) nor economical supports are included in our usage of the lockdown.

We divided population into young (15–49 years old), middle (50–64 years old) and old (more than 64-year-old) populations and ignored child age group (0–14 years old) since age was critical factor for contact rates and mortality, and also for simplicity of simulation settings [18]. 10 vaccine allocation strategies were compared in the present study. One scenario was equal allocation for all age groups. 6 scenarios were precise prioritization for 3 generations. For example, old generations were targeted at first. If all old individuals received vaccination, the next target was middle age group. After that, young generations received vaccination. The other 3 scenarios were partial prioritization strategies. One age group was targeted at first and the rest of vaccines were allocated equally to the other two age groups.

Regarding lockdown strategies, it was assumed that the lockdown was performed only once during each simulation since there was a small chance of resurgence of COVID-19 at the latter phase of the simulation due to vaccination. To balance economy with infection control, we varied three parameters related with the lockdown: start timing, $t_s$, intensity for age group $j$, $L_j$, and length, $T_L$, of the lockdown. In our main analyses, the lockdown was equally imposed to all generations so that $L_j = L$ for any age group. Simulations with different intensities of the lockdown between generations were performed as a sensitivity analysis.

Loss of total production value due to the lockdown was defined as economic loss, $E$, expressed as a proportion to the non-lockdown situation and calculated as

$$E = \frac{T_L}{365} \theta \left( \frac{\sum_j w_j N_j L_j}{\sum_j w_j N_j} \right)^{1+\phi},$$ (1)

where $w_j$ and $N_j$ was an average production value and the number of populations for age group $j$, respectively. $\theta$ was defined as lockdown efficacy and we assumed 25% of population would not be affected by the lockdown. We did not consider loss of production caused by any other countermeasures. The relationship between lockdown intensities and economic loss was assumed to be linear ($\phi = 0$) for the main results. Also, non-linear relationships expressed as a power function were applied. This non-linearity came from the discussion in [19]. In this setting, mild countermeasures against infectious diseases such as teleworking could be taken without significant economic loss, but stronger measures such as the closure of companies to achieve stronger behavioral restraint requirements would result in greater economic loss. Simulations with values of one, two or three for $\phi$ were also performed for the case of the lockdown imposed equally to each age group.

In addition to analysis with 1.3 of basic reproduction number, $R_0$, we performed sensitivity analysis varying the basic reproduction number. Also, we performed the same analysis with age group specific lockdown intensity, $L_j$, varying $R_0$.

We estimated parameters, $L_j$ and $t_s$, to minimize the cumulative number of deaths. $T_L$ were calculated from $L_j$ and $E$ for each step. Ordinal differential equation was solved by Explicit Runge-Kutta method of order 5(4) [20] and optimization was done with differential evolution implemented in a Python package, Scipy version 1.5.3 [21, 22].

## SEIR model scheme

Equations of our SEIR model in the present study are shown as

$$\frac{dS_{j,u}(t)}{dt} = -\lambda_j(t) S_{j,u}(t) - V_{j,u}(t) S_{j,u}(t),$$ (2)

$$\frac{dE_{j,u}(t)}{dt} = \lambda_j(t) S_{j,u}(t) - \tau E_{j,u}(t) - V_{j,u}(t) E_{j,u}(t),$$ (3)

$$\frac{dI_{j,u}(t)}{dt} = \tau E_{j,u}(t) - \gamma I_{j,u}(t) - V_{j,u}(t) I_{j,u}(t),$$ (4)

$$\frac{dR_{j,u}(t)}{dt} = \gamma \left( 1 - IFR_j \right) I_{j,u}(t) - V_{j,u}(t) R_{j,u}(t),$$ (5)

$$\frac{dD_{j,u}(t)}{dt} = \gamma IFR_j I_{j,u}(t) - V_{j,u}(t) D_{j,u}(t),$$ (6)

$$\frac{dS_{j,v}(t)}{dt} = -(1 - VE_\lambda)\lambda_j(t)S_{j,v}(t) + V_{j,u}(t)S_{j,u}(t), \tag{7}$$

$$\frac{dE_{j,v}(t)}{dt} = (1 - VE_\lambda)\lambda_j(t)S_{j,v}(t) - \tau E_{j,v}(t) + V_{j,u}(t)E_{j,u}(t), \tag{8}$$

$$\frac{dI_{j,v}(t)}{dt} = \tau E_{j,v}(t) - \gamma I_{j,v}(t) + V_{j,u}(t)I_{j,u}(t), \tag{9}$$

$$\frac{dR_{j,v}(t)}{dt} = \gamma\Big(1 - (1 - VE_d)IFR_j\Big)I_{j,v}(t) + V_{j,u}(t)R_{j,u}(t), \tag{10}$$

$$\frac{dD_{j,v}(t)}{dt} = \gamma(1 - VE_d)IFR_jI_{j,v}(t) + V_{j,u}(t)D_{j,u}(t), \tag{11}$$

where

$$V_{j,u}(t) = \frac{Np_j(t)}{T_{vac}}\frac{1}{N_{j,u}(t)}, \tag{12}$$

and semantics of them are depicted in Fig 1.

Our model consisted of contact matrix, infection fatality ratio (IFR) and vaccinated and unvaccinated compartments for simulating infection dynamics of age-stratified populations with vaccination. We refers $S_{j,u}$ and $S_{j,v}$ as unvaccinated and vaccinated susceptible populations of age group $j$. Exposed, infected and recovered populations are similarly defined. Vaccines were distributed at a rate of $N/T_{vac}$ for the population and $Np_j/T_{vac}N_{j,u}$ for each compartment where $N$ is total number of the population, $T_{vac}$ is the whole interval of vaccination, and $N_{j,u}$ is total number of unvaccinated population. Vaccine allocation percentage for age group $j$, $p_j$ is time-varying parameters taking between 0 to 1 determined by vaccine allocation strategies. For old-other strategies, a mean vaccine allocation percentage for old generation, $p_o$, takes 1 till the end of their vaccination and the others are 0. After that, mean vaccine allocation

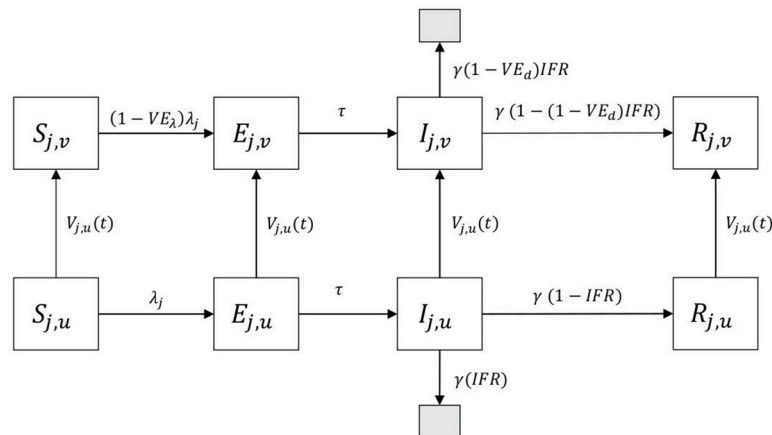

**Fig 1. Schematic of SEIR model composed of vaccinated and non-vaccinated compartments.** Subscript $j$ represents young, middle, or old age group. Black shaded box represents death status.

percentages for young, $p_y$, and middle generations, $p_m$, take percentage of population among two to distribute vaccine according to population size until the whole population is vaccinated. Vaccine effect is assumed to be a leaky one, meaning that vaccinated populations were less likely, not completely, to be infected compared to unvaccinated individuals. Infected and recovered individuals also received vaccination in our model. There are three reasons for this. One reason is that insufficient immunity has been controversial now [23], and that vaccination would contribute to reduce reinfection. The third reason is that it is difficult for asymptomatic cases to be separated from susceptible ones without tests. Death cases are also assumed to be vaccinated in the model to carry out a predetermined vaccination plan.

## Force of infection and basic reproduction number

Force of infection for age group $j$, $\lambda_j$, is an inflow rate from susceptible population to exposed population. Taking lockdown effect into account, force of infection for age group $j$ is derived as

$$\lambda_j = \frac{u}{N}\left(1 - \theta L_j(t)\right)\sum_k \rho_{jk}(1 - \theta L_k(t))(I_k(t) + I_{k,v}(t)), \tag{13}$$

where $u$ is a successful transmission rate given one contact, and $\rho_{jk}$ is a contact rate for age group $j$ with age group $k$.

The basic reproduction number was calculated from absolute of the dominant eigen value of the matrix $M$, which was the product of a matrix $C$ and a diagonal matrix of which element was $u/\gamma$ where $1/\gamma$ was infectiousness period. The matrix $C$ represents contact matrix weighted by population size which elements were written as $\rho_{jk} N_k$. This derivation method of $R_0$ was described in [24].

## Parameter specification

Parameters used in this study are summarized in Table 1. Most parameters came from resources in Japan in order to perform simulation under the Japanese setting. We had to modify several original data to be compatible with our model, which were contact rates, IFR and production values.

Table 1. Summary of parameters used in the simulations.

| Symbols | Descriptions | Values | References |
|---|---|---|---|
| $\tau$ | 1/ Latent period (/day). | 1/3 | [25] |
| $\gamma$ | 1/ Infectiousness period (/day). | 1/5 | [26] |
| $u$ | Transmission rate given one contact. | 0.05882 when $R_0 = 1.5$ | Calculation |
| $\rho_{jk}$ | Contact matrix for age group $j$ with age group $k$ [†]. | [[10.482, 1.567, 0.332], [7.250, 3.506, 0.879], [3.105, 1.789, 2.926]] | [27] |
| $IFR_j$ | Infection fatality ratio for age group $i$ (%) [†] | [0.030, 0.295, 4.893] | [28] |
| $N_j$ | Population for age group $j$ ($\times 10^3$) [†]. | [50557, 23987, 36155] | [29] |
| $[E(0), I(0), R(0)]$ | Initial number for E, I and R compartments[*]. | [9540, 18685, 358145] | [30] |
| $\theta$ | Lockdown efficiency. | 0.75 | Assumption |
| $w_j$ | Production value for age group $j$ ($\times 10^3$ yen) [†]. | [3742, 4474, 812] | [31, 32] |
| $VE_\lambda$ | Vaccine efficacy for transmission. | 0.95 | [33] |
| $VE_d$ | Vaccine efficacy for death. | 0.84 | [34] |

[†] The order of values listed is young, middle and old age group.

[*] These values were assigned to each age group proportional to population size.

To truncate the original data to the values used in the model, we denote $x_{5i,5j}$ as contact rates of a person among *5i* to *5(i+1) -1* age interval with a person among *5j* to *5(j+1) -1* age interval and $N_{5i}$ represents population size for *5i* to *5(i+1) -1* years old since original data had 5-year interval values. We combined rows for each age interval and took weighted summarization for columns by population size. For example, contact rate of young group with middle group, $\rho_{ym}$, was calculated as

$$\rho_{ym} = \frac{\sum_{i=3}^{9} N_{5i} \sum_{j=10}^{12} x_{5i,5j}}{\sum_{i=3}^{9} N_{5i}}, \tag{14}$$

where contact rates for young with middle age group were summed over from 50–54 to 60–64 age interval and were weighted by each age interval from 15–19 to 45–49 age interval. Similar calculation was done for other age group pairs, noting that we ignored population from 0 to 14 years old. The contact rates of the reference paper [27] did not have the ones of age more than 79 years old, so that we copied rows and columns of 75–79 years old diagonally. We multiplied 0.9 for these rows and columns since we considered older populations had lower contact rate than that of 75–79 years.

The reference of IFR [28] showed that the relationship between IFR and age was exponential. Since we were able to calculate infection fatality ratio at any age using this relationship, IFR of each age group was calculated as weighted means of IFR for each 5-age interval. For example, the equation for young age group was derived as

$$IFR_y = \frac{\sum_{i=3}^{9} N_{5i} IFR_{5i+2}}{\sum_{i=3}^{9} N_{5i}}, \tag{15}$$

where $IFR_{5i+2}$ represents the infection fatality rate at age *5i +2*, which is a median of each age interval. For old age group, more than 80 years old were grouped as one and we chose IFR at 85 years old for this group.

Production value was intended to reflect production values by workers. We obtained values of salary for each 5-age intervals, and weighted summaries by population size were used for production value for each age group, $w_j$.

We set the initial number of each SEIR component as the status on 1st February 2021 in Japan. We set initial value for $E_j$ as sum of newly reported cases during 29th to 31st January 2021 multiplied by a percentage of population size for age group *j* and for $I_j$ and $R_j$ as sum of ones during 23rd to 28th January 2021 and the cumulative number of cases till 22nd January 2021, respectively, multiplied by the same percentage. Actual values can be seen in Table 1.

## Results

Fig 2 illustrates the relationship between economic loss and lockdown intensity and length values. x and y axis represent intensity and length of the lockdown and z axis represents the cumulative number of deaths at the end of the simulation. Optimization of parameters ($T_L$, $L$ and $t_s$) for the lockdown imposing equally to each age group was done along each line drawn by constraint of predefined economic loss value, $E$. The start timing of the lockdown, $t_s$, was estimated to be $0^{th}$ day for almost all simulations. $t_s$ was estimated to be more than $0^{th}$ day (sometimes around $70^{th}$ day) when $\phi$ took a value of 0, $R_0$ was higher, economic losses were lower, and the strategies were the middle first ones, the old first ones, or the equal strategy. In the case of a value of $\phi$ more than zero, $t_s$ was estimated to be $0^{th}$ day for all cases. Then, we omitted the value of $t_s$ from the following figures.

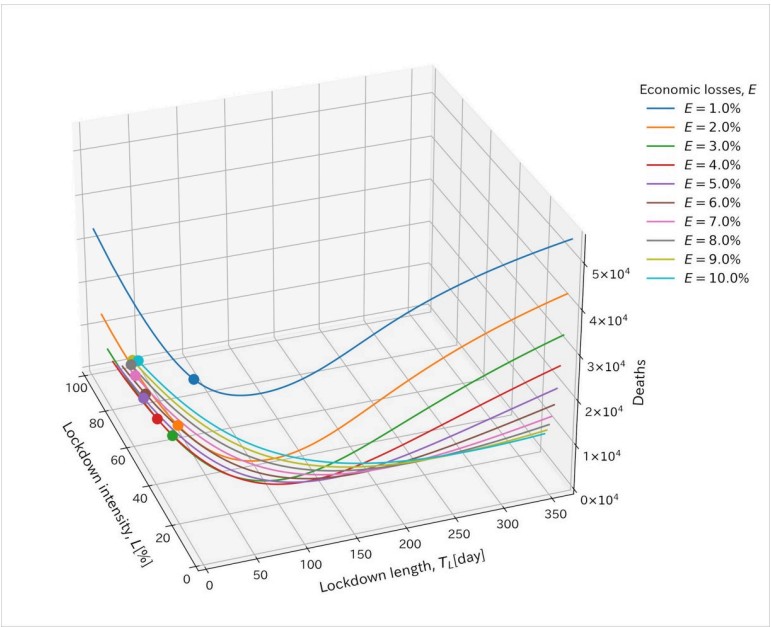

**Fig 2. Illustration of relationship between economic loss and lockdown intensity and length values.** x and y axis represent lockdown intensity and length values. z axis represents the cumulative number of deaths at the end of the simulations. Each line was drawn with the same volume of economic loss, $E$. Points show the minimum cumulative number of deaths for each line. Values at these points were used for the optimized parameters and shown in the following figures.

Fig 3 summarizes the cumulative number of deaths and infected population at the end of the simulations under 10 vaccine allocation strategies for each acceptable economic loss, together with the lockdown intensity and length giving optimal results (see S1 Table for detail).

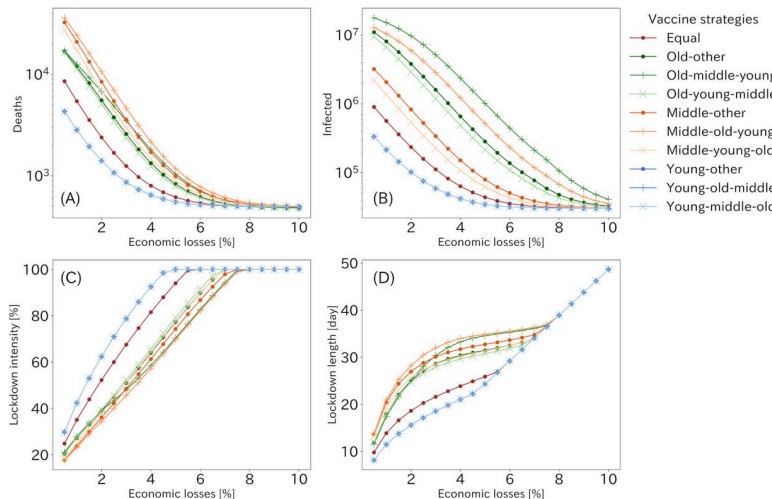

**Fig 3. The optimized results and parameters for R0 = 1.3 and $\phi$ = 0 when the lockdown was imposed equally to each generation.** The optimized results of the cumulative number of deaths (A) and infected (B) populations for each economic loss. (C) and (D) represents the lockdown intensity and length, which value pattern produced the results of (A) and (B) for each economic loss. Although there are only eight lines visible in (A) and (B), three lines of young first vaccination strategies have overlain each other. The order of age groups on the label shows vaccine allocation strategies.

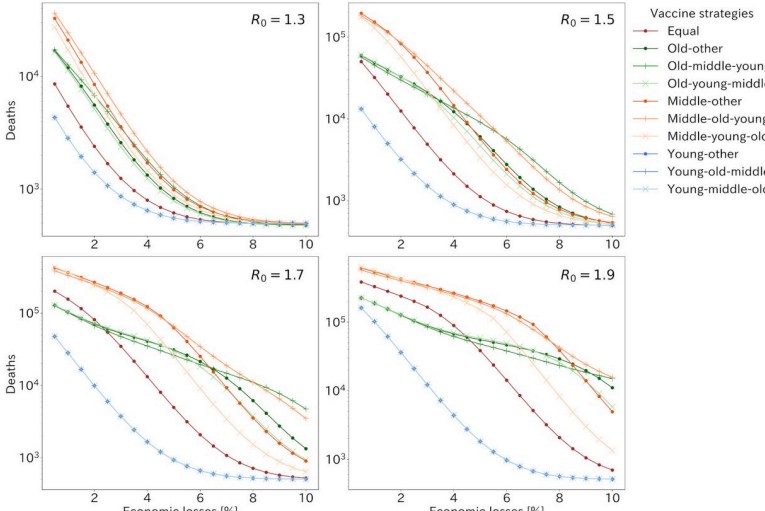

**Fig 4. The cumulative number of deaths with R0 taking 1.3, 1.5, 1.7 and 1.9 and $\phi$ = 0 when the lockdown was imposed equally to each generation.** Each line shows vaccine allocation strategies. Three lines of young first vaccination strategies have overlain each other.

The lockdown was equally imposed to all generations, the basic reproduction number, $R_0$, was set to be 1.3 and $\phi$ took zero. Overall, three young first strategies were the most effective in reducing the number of deaths and infections for most of the cases. The fourth effective strategy would be one that distributed vaccine equally to all age groups, especially in lower acceptable economic loss. The strong lockdown with short interval was preferable for young first strategies since young population had high contact rates. Restricting and vaccinating this population simultaneously leaded to lower reproduction number.

The cumulative numbers of the deaths for all strategies under 4 patterns of the basic reproduction number when $\phi$ took zero are shown in Fig 4. The best strategies were young first ones in terms of the cumulative number of deaths or infected population regardless of $R_0$ except the strong lockdown situation. In addition, the strategy distributing vaccine equally to all age groups was the fourth effective strategy in most sizes of economic losses under 1.5 of $R_0$. However, when $R_0$ was more than around 1.5, in situations where the lockdown was mild and caused little economic loss, three old first strategies were more effective than those of vaccinating all generations equally in terms of deaths. When comparing strategies prioritizing vaccination to the old or middle age group, the old first strategies were more effective when the lockdown caused little economic loss, while the middle first strategies were more effective when there was much economic loss due to the lockdown. The differences in the effectiveness of the middle and old first strategies were more apparent for larger values of $R_0$.

The heatmap of the best strategies in terms of the cumulative number of deaths among the old strategies showed that early vaccination for the young (specifically the young-old-middle strategy) was most effective in reducing the number of deaths when $\phi$ was zero (Fig 5). However, as $\phi$ increased, the old first strategies were best strategies in case of low $R_0$ and higher acceptable economic losses. Particularly, for the case where $R_0$ was small and the acceptable economic loss was large, the old first strategies was clearly best strategies among all when $\phi$ took three (S1 Fig).

As Japan has already decided to vaccinate to the old age groups first, we compared three old first strategies in order to clarify which order of vaccination was better after the old age group was vaccinated. Fig 6 summarizes the best old first strategy among three strategies, old-

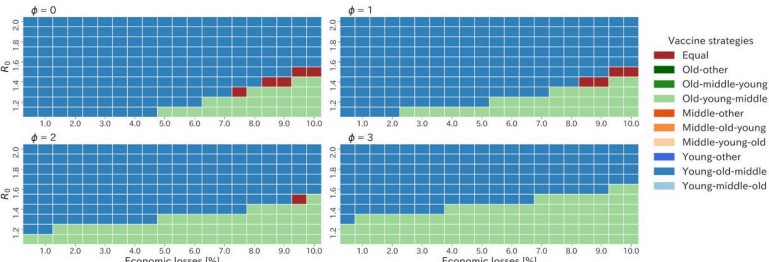

**Fig 5. Heatmap of the best strategies in terms of the cumulative number of deaths among all strategies varying $\phi$, R0 and acceptable economic losses.**

middle-young, old-young-middle, and old-other strategies, varying $\phi$ and $R_0$ (see S2 Table for detail). When $R_0$ was below 1.4 and $\phi$ took zero, the old-young-middle strategy was most effective in reducing deaths under almost all the economic loss patterns, whereas under larger $R_0$, the old-middle-young strategy started to excel greater than the other strategies for low acceptable economic loss. For the value of $\phi$ more than zero, the old-young-middle strategy was the best for the almost all the cases. Especially, if $\phi$ took three, the old-young-middle strategy was best for all the settings.

We also performed simulations under settings of the lockdown being imposed with different intensities to each age group for 1.3 of $R_0$ (S2 and S3 Figs) and varying $R_0$ (S4 and S5 Figs) when $\phi$ took zero. Over all trends of the cumulative number of deaths and infected population were similar as when the lockdown was equally imposed. One exception was when $R_0$ took 2.0 and economic loss was 0.5% and the best strategy was old-middle-young strategy (S5 Fig). To achieve optimal strategies, the strong lockdown to young and old age group was preferable compared to middle age group.

## Discussion

We explored the better vaccine allocation and intensive countermeasure strategies to balance economic sustainability with infection control against COVID-19 in Japan. The young first strategies (specifically the young-old-middle strategy) were better than any other strategies in lower acceptable economic loss and moderate to higher $R_0$ when a linear relationship between lockdown intensity and acceptable economic losses ($\phi = 0$) was assumed (Fig 5). If we applied a non-linear relationship expressed as a power function, which implied a large potential for reducing contact without economic loss, the old first strategy (specifically the old-young-middle strategy) became the best when lower $R_0$. These results were obtained because the spread of infection was suppressed only by lockdown, and as a result, the effect of early vaccination of the young on the prevention of the spread of infection was diminished, and instead the effect of early vaccination of the elderly on the reduction of mortality was greatly contributed. These results indicated a high potential of remote work when prioritizing vaccination for the old generation.

If we focused on the old first strategies, as the Japan government has decided to vaccinate the old populations first, the old-young-middle strategy was the best for almost all the cases for $\phi$ more than zero (Fig 6). These results came from the fact that young age group had the highest contact rates for all age groups [25] and was more likely to transmit infection so that vaccinating young populations at an early stage would prevent them from transmitting infection and enable population to reach herd immunity at the early stage, resulting in the containment of the epidemic and reduction of deaths. Also, the non-linear relationship would contribute to

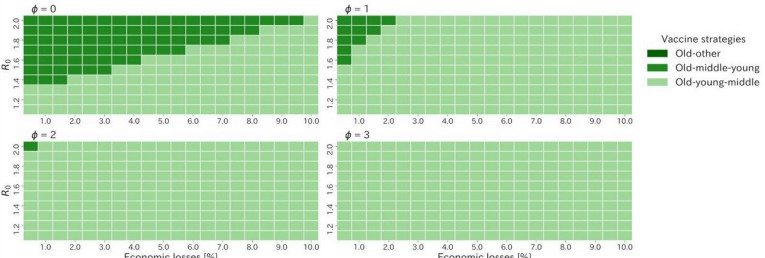

**Fig 6. Heatmap of the best strategies in terms of the cumulative number of deaths among the old first strategies varying ϕ, R0 and acceptable economic losses.**

containments of the epidemics and the role of vaccination came to be a containment of the epidemic.

Our results implied that the best strategy minimizing deaths was changed according to the value of $R_0$ and how countermeasures causing economical damage was imposed. $R_0$ of COVID-19 was estimated as more than 2 at early stage of the epidemic [35, 36]. However, effective case tracing [26, 37] and much public awareness reduced transmissibility and some study handling with negative values of serial interval reported lower value of $R_0$ [38]. Also, our usage of a lockdown does not include countermeasures without economical damage, which will lead to lower $R_0$. However, a new SARS-CoV-2 variant, VOC 202012/01 was reported to be 43–90% higher transmissibility than preexisting variants [39, 40] and has been appeared also in Japan. It indicates higher $R_0$ setting is better for our results. As shown in Fig 4, as $R_0$ increased, number of deaths increased exponentially, so that strategies should be chosen by assuming the worst.

Since we used relative volume of population size and salary for each age group for calculation of economic loss, $E$, comparison with other metrics can be done. The annual GDP growth rates for 2019 and 2020 were 0.3% and -4.8%, respectively [16]. If we focus on the quarterly GDP growth in 2020, these values are -0.6%, -8.3%, 5.3% and 2.8% from the earliest to the latest. If this volume of decline is totally caused from countermeasures and contributes to reduce transmissibility, sufficient decline can be achieved. However, there are many measures to reduce transmissibility without economical damage so that raising each person's awareness of infection control is important.

While our analysis assumed that the entire population would eventually be vaccinated in one year, this assumption would not be accurate in terms of vaccination rate and final percentage of vaccination. Also, vaccination efficacy for transmissibility and deaths were not accurately evaluated [34] and how long its immunity sustains and how protective for new strains it is remains unknown.

As for a lockdown, its intensity in the present study was assumed to be precisely controlled by policy makers but in practice the actual extent of the contact reduction may not be obvious until a lockdown is in practice. Although the one-time lockdown assumption was adopted in this study, this implementation should be caution in the real-world settings. The simple SEIR model study [19] showed constant strong lockdown strategy was not optimal if the vaccination was started at the end of the simulation. The gradual relaxation strategy was preferred, and this strategy was conducted in the UK and Israel during vaccination. For the lockdown with age-specific intensities, the results were not much changed from the lockdown with the same intensity to all age groups (S5 Fig). The one exceptional result in S5 Fig when $R_0$ took 2.0 and economic loss was 0.5% indicated old-first strategy started to beat other strategies in case of higher $R_0$ with weak lockdown.

There are several limitations to our study. First, although the results of this study relied highly on parameters, especially contact rates, the contact rates referred to in the present study were based on data from [27], which did not take into account behavioral changes brought about by the COVID-19 pandemic. Second, our analysis excluded children aged from 0 to 15 years old. Several studies suggested children were less likely to contribute to epidemics [25, 40–42], the number of children cases has been increasing due to a new variant of SARS-CoV-2. Third, our model of economic loss considered only the impact of direct behavioral restraint due to the lockdown, and did not take into account the value that infectious disease victims were expected to produce in the future, which had been taken into account in several previous studies about economic losses by COVID-19 pandemic [18, 43]. Finally, since the calculations in this study were based on the population structure and contact rates in Japan, the results obtained in this study are not directly applicable to the situation of foreign countries.

## Supporting information

**S1 Fig. The cumulative number of deaths with $R_0$ taking 1.3, 1.5, 1.7 and 1.9 and $\phi$ taking 1, 2, and 3 when the lockdown was imposed equally to each generation.** Each line shows vaccine allocation strategies. Three lines of young first vaccination strategies have overlain each other.
(TIF)

**S2 Fig. The optimized results for $R_0$ = 1.3 when the lockdown was imposed with different intensities to each age group.** The optimized results of the cumulative number of deaths and infected population for each economic loss is presented. Three lines of young first vaccination strategies have overlain each other. The order of age groups on the label shows vaccine allocation strategies.
(TIF)

**S3 Fig. The optimized parameters for $R_0$ = 1.3 when the lockdown was imposed with different intensities to each age group.** Each figure block contains lockdown intensities for young (blue), middle (yellow) and old (green) age group and lockdown length (red). It is noted that y axis represented lockdown intensity and lockdown length.
(TIF)

**S4 Fig. The cumulative number of deaths with $R_0$ taking 1.3, 1.5, 1.7 and 1.9, and $\phi$ = 0 when the lockdown was imposed with different intensities to each age group.** Each line shows vaccine allocation strategies. Three lines of young first vaccination strategies have overlain each other.
(TIF)

**S5 Fig. Heatmap of the best strategy in terms of the cumulative number of deaths among all strategies varying $R_0$ and acceptable economic losses when the lockdown was imposed with different intensities to each age group.**
(TIF)

**S1 Table. Numerical results for the beset strategy among all to minimize the cumulative number of deaths imposing same intensity of the lockdown to all age groups.**
(DOCX)

**S2 Table. Numerical results for the beset strategy among the three old first strategies to minimize the cumulative number of deaths imposing same intensity of the lockdown to all**

**age groups.**
(DOCX)

## Acknowledgments

We thank for Yusuke Asai for valuable technical advice and useful discussions.

## Author Contributions

**Conceptualization:** Satoshi Sunohara, Toshiaki Asakura.

**Data curation:** Toshiaki Asakura.

**Formal analysis:** Satoshi Sunohara, Toshiaki Asakura.

**Funding acquisition:** Akiko Tamakoshi.

**Investigation:** Satoshi Sunohara, Toshiaki Asakura, Shun Ozawa, Satoshi Oshima, Daigo Yamauchi.

**Methodology:** Satoshi Sunohara, Toshiaki Asakura.

**Project administration:** Takashi Kimura, Akiko Tamakoshi.

**Resources:** Satoshi Sunohara, Toshiaki Asakura, Shun Ozawa.

**Software:** Satoshi Sunohara, Toshiaki Asakura.

**Supervision:** Takashi Kimura, Akiko Tamakoshi.

**Validation:** Toshiaki Asakura.

**Visualization:** Satoshi Sunohara, Toshiaki Asakura.

**Writing – original draft:** Satoshi Sunohara, Toshiaki Asakura, Takashi Kimura, Shun Ozawa, Satoshi Oshima, Daigo Yamauchi.

**Writing – review & editing:** Takashi Kimura, Akiko Tamakoshi.

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
