## [Decision Letter · Decision Letter 0]

19 May 2021

PONE-D-21-12208

Effective vaccine allocation strategies, balancing economy with infection control against COVID-19 in Japan

PLOS ONE

Dear Dr. Kimura,

Thank you for submitting your manuscript to PLOS ONE. After careful consideration, we feel that it has merit but does not fully meet PLOS ONE’s publication criteria as it currently stands. Therefore, we invite you to submit a revised version of the manuscript that addresses the points raised during the review process.

We look forward to receiving your revised manuscript.

Kind regards,

Martial L Ndeffo Mbah, Ph.D

Academic Editor

PLOS ONE

Journal Requirements:

Reviewers' comments:

Reviewer's Responses to Questions

**Comments to the Author**

1. Is the manuscript technically sound, and do the data support the conclusions?

Reviewer #1: Partly

Reviewer #2: Yes

Reviewer #3: Partly

2. Has the statistical analysis been performed appropriately and rigorously? 

Reviewer #1: Yes

Reviewer #2: Yes

Reviewer #3: N/A

3. Have the authors made all data underlying the findings in their manuscript fully available?

Reviewer #1: Yes

Reviewer #2: Yes

Reviewer #3: Yes

4. Is the manuscript presented in an intelligible fashion and written in standard English?

Reviewer #1: Yes

Reviewer #2: Yes

Reviewer #3: Yes

5. Review Comments to the Author

Reviewer #1: The paper is on "Effective vaccine allocation strategies, balancing economy with infection control against COVID-19 in Japan". There were several articles published on that topic this year, and I do not think that this paper enters the top 20%. There is nothing really new here. Here are some comments

1) (42) "COVID-19 has affected 42 tens of millions of people" it might depend on what "affect" means but if we include lockdowns for instance, it would be more "billions"

2) (58) the grammar "vaccination in healthcare workers" is odd, but I am not a native English spearker

3) (88) "lockdown was performed only once" is that a relevent assumption ? so far, many countries have faced second or third wave, so it would make sense to assume that multiple lockdown is possible. Furhermore I have major concern about the design of the lockdown : it is constant, over a given period of time, and uniform over the classes (same theta). Why ? In equation (1) the economic loss is linear... can't we assume that some people can work from home ? can't we assume that E is not linear in theta ? or N ? These assumptions are extremely strong, and not enough discuss. The problem is that all conclusions rely on those assumptions

4) SEIR model (105) is that relevant for a COVID-19 type of pandemic ? what about asymptomatic ? that was the main challenge at first, and a strong motivation for lockdowns early 2020

5) (128) again, things are uniform over classes... aren't there are more interactions within age classes ? Here having a similar theta is weird

Reviewer #2: Reviewer Report

(Recommendation)

I recommend the research article. However, it would be appropriate for the authors to consider some recommendations.

1. Summary of the investigation:

The article uses the SEIR model. Apply an optimization method with the differential evolution method and use a proven Python algorithm. The research question: What is the most appropriate vaccination strategy based on economic cost? and of course, it assumes other variables or parameters that the literature and evidence review suggests. In this context, it seeks to balance the economy and infection control in Japan.

One of the greatest strengths is in having a proven methodology that is better than other existing ones. But as a weakness is that no appendix shows how the theoretical-mathematical model derives, and secondly, to be able to replicate the results in Python. Of course, this situation does not affect the results.

The main result of the study is that the effective vaccination strategy that prioritizes the younger generation outperformed the other strategies in terms of deaths, explained by the contact rate and therefore the transmission.

2. Evidence and examples:

First, a review of the literature on the current situation of the Vaccine against the Pandemic would be important. It could expand the description of the state of the art of public health policies that have been carried out and compare not only at the level of Japan but at least at the Asian or the Asia Pacific and/or Global level.

Second, review of the empirical evidence that can be presented (Results) and compared (Discussion) in the present study. It is important to compare the results with others in the region, at the level of Asia, Asia-Pacific, at the level of developed countries or the world.

Reviewer #3: Please see an attachment file.

Please see an attachment file.

Please see an attachment file.

Please see an attachment file.

Please see an attachment file.

Please see an attachment file.

Please see an attachment file.

6. PLOS authors have the option to publish the peer review history of their article (what does this mean?). If published, this will include your full peer review and any attached files.

Reviewer #1: No

Reviewer #2: No

Reviewer #3: **Yes: **Tsuyoshi HONDOU

---

## [Author Response · Author response to Decision Letter 0]

6 Aug 2021

We are most grateful to you and the reviewers for the comments on our manuscript.

See the file uploaded to response to reviewers.

---

## [Decision Letter · Decision Letter 1]

24 Aug 2021

Effective vaccine allocation strategies, balancing economy with infection control against COVID-19 in Japan

PONE-D-21-12208R1

Dear Dr. Kimura,

We’re pleased to inform you that your manuscript has been judged scientifically suitable for publication and will be formally accepted for publication once it meets all outstanding technical requirements.

Kind regards,

Martial L Ndeffo Mbah, Ph.D

Academic Editor

PLOS ONE

Additional Editor Comments (optional):

Reviewers' comments:

Reviewer's Responses to Questions

**Comments to the Author**

1. If the authors have adequately addressed your comments raised in a previous round of review and you feel that this manuscript is now acceptable for publication, you may indicate that here to bypass the “Comments to the Author” section, enter your conflict of interest statement in the “Confidential to Editor” section, and submit your "Accept" recommendation.

Reviewer #1: All comments have been addressed

Reviewer #2: All comments have been addressed

2. Is the manuscript technically sound, and do the data support the conclusions?

Reviewer #1: Yes

Reviewer #2: Yes

3. Has the statistical analysis been performed appropriately and rigorously? 

Reviewer #1: N/A

Reviewer #2: Yes

4. Have the authors made all data underlying the findings in their manuscript fully available?

Reviewer #1: Yes

Reviewer #2: Yes

5. Is the manuscript presented in an intelligible fashion and written in standard English?

Reviewer #1: Yes

Reviewer #2: Yes

6. Review Comments to the Author

Reviewer #1: The paper has been substentialy improved. I still do no see real innovations in the paper (compared with several papers published recently on the same topic), but since the editor and the two other referees think that it is relevant that such a paper could be published in PLOS ONE, I will not go against it

Reviewer #2: The authors respond appropriately to the reviewers' suggestions.

The authors use and extend the SEIR model. They apply an optimization method with the differential evolution method and use a Python algorithm.

The authors manage to add the observations by making an article more suitable for publication according to the journal Plos-One.

7. PLOS authors have the option to publish the peer review history of their article (what does this mean?). If published, this will include your full peer review and any attached files.

Reviewer #1: No

Reviewer #2: No

---

## [Editor Report · Acceptance letter]

26 Aug 2021

PONE-D-21-12208R1 

Effective vaccine allocation strategies, balancing economy with infection control against COVID-19 in Japan 

Dear Dr. Kimura:

I'm pleased to inform you that your manuscript has been deemed suitable for publication in PLOS ONE. Congratulations! Your manuscript is now with our production department. 

Kind regards, 

on behalf of

Dr. Martial L Ndeffo Mbah 

Academic Editor

PLOS ONE